# Predictive role of pretreatment skeletal muscle mass index for long-term survival of bladder cancer patients: A meta-analysis

Qian Yuan[1,2], Jianrong Hu[1,2], Feng Yuan[1,2], Jingjing An [1,2]*

1 Department of Anesthesiology, West China Hospital, Sichuan University, Chengdu, China, 2 West China School of Nursing, Sichuan University, Chengdu, Sichuan, China

* anjingjing2022@163.com

## Abstract

### Purpose

To identify the predictive role of pretreatment skeletal muscle mass index (SMI) for long-term survival of bladder cancer patients.

### Methods

Several databases were searched for studies investigating the relationship between pretreatment SMI and prognosis in bladder cancer. The overall survival (OS) and cancer-specific survival (CSS) were defined as primary and secondary outcomes, respectively. Hazard ratios (HRs) and 95% confidence intervals (CIs) were combined.

### Results

Nine studies involving 1476 cases were included. The results demonstrated that a lower pretreatment SMI was significantly related to poorer OS (HR = 1.56, 95% CI: 1.33–1.82, P<0.001) and subgroup analysis based on thresholds of SMI revealed similar results. Besides, pretreatment SMI was also obviously related to CSS (HR = 1.75, 95% CI: 1.36–2.25, P<0.001).

### Conclusion

Lower pretreatment SMI was associated with worse long-term survival of bladder cancer patients.

## Introduction

Bladder cancer remains one of the most common urinary malignancies and mainly occurs in the elderly patients [1]. According to the latest cancer data, there was 570,000 new cases in 2020 all over the world, with the tenth morbidity [2]. For male patients, 440,000 bladder cancer cases occur with the sixth morbidity and 160,000 patients died with the ninth mortality among

**Data Availability Statement:** All data used in this meta-analysis are presented in the paper.

**Funding:** The author(s) received no specific funding for this work.

**Competing interests:** The authors have declared that no competing interests exist.

all cancers in 2020 [2]. Besides, the incidence tends to increase gradually, which causes a certain tumor burden to the society. Up to now, the prognosis of bladder cancer is still poor despite advances in surgical technologies and chemotherapy [3, 4].

Increasing evidence has demonstrated that tumor progression and prognosis depend not only on the biological aggressiveness of the tumor but also on the host's response to the tumor. Host factors such as the nutritional status and local or systemic inflammation response are also important indicators of clinical treatment [5]. Systemic inflammation response index (SII) has been verified to be significantly associated with treatment response and survival of bladder cancer patients [6, 7]. Loss of weight and body mass index (BMI) are usually applied to evaluate the nutritional status and cachexia in cancer patients, but these indexes only reflect the total body composition and do not distinguish the proportion and change of fat and muscle mass. Actually, the muscle mass is significantly related to the overall body condition and nutritional status of cancer patients [8, 9].

Many studies have manifested that sarcopenia could reflect potential malnutrition and weakness caused by cancers and skeletal muscle mass index (SMI) is the most authoritative indicator to evaluate the presence or absence of sarcopenia in cancer patients [10, 11]. SMI is calculated by dividing the total area of all skeletal muscles, including the psoas major muscle, erector spinae muscle, quadratus lumborum muscle, transverse muscle of abdomen, obliquus externus abdominis and obliquus internus abdominis, in the third lumbar level of CT images by the square of height [12]. Up to now, the association of pretreatment SMI with long-term survival has been verified by meta-analyses in several types of cancers like the lung cancer [12, 13]. However, the prognostic value of pretreatment SMI in bladder cancer remains unclear now.

Therefore, the aim of this meta-analysis was to identify predictive role of pretreatment SMI for long-term survival of bladder cancer patients.

## Materials and methods

This meta-analysis was performed according to the Preferred Reporting Items for Systematic Reviews and Meta-Analysis (PRISMA 2020) checklist [14]. The detailed checklist information was presented in the S1 File.

### Literature search

The PubMed, EMBASE, WOS and CNKI database were searched up to September 21, 2022. Terms used during the literature search are as follows: skeletal muscle mass index, SMI, bladder, tumor, cancer, neoplasm, carcinoma, survival, prognostic and prognosis. Search strategy was as follows: (skeletal muscle mass index OR SMI) AND bladder AND (tumor OR cancer OR neoplasm OR carcinoma) AND (survival OR prognostic OR prognosis). Besides, the free texts and MeSH terms were used.

### Inclusion criteria

The inclusion criteria included: 1) patients were diagnosed with primary bladder cancer; 2) SMI was calculated according to the CT images of the third lumbar vertebra as previously reported [15]; 3) the SMI values were obtained before anti-tumor therapy such as the surgery and chemoradiotherapy; 4) patients were divided into two groups according to values of SMI and long-term survival representing as the overall survival (OS) and cancer-specific survival (CSS) were compared; 5) hazard ratios (HRs) and 95% confidence intervals (CIs) were provided in the articles.

## Exclusion criteria

The exclusion criteria included: 1) letters, editorials, case reports, reviews or animal trials; 2) duplicated or overlapped data; 3) insufficient information for methodological quality assessment.

## Data extraction

Data were collected from included studies: the name of first author, publication year, country, sample size, tumor-node-metastasis (TNM) stage, treatment (surgery or non-surgery), cutoff value of SMI, endpoint, HR and 95% CI.

## Methodological quality assessment

Methodological quality was evaluated according to Newcastle-Ottawa Scale (NOS) score due to the retrospective nature of study design [16]. Studies with a NOS score ≥6 were regarded as high-quality studies.

## Statistical analysis

Statistical analysis was conducted by STATA 15.0 software. HRs with 95% CIs were combined to assess the relationship between pretreatment SMI and prognosis of bladder cancer patients. The heterogeneity among included studies was evaluated by $I^2$ statistics and Q test. When significant heterogeneity was observed representing as $I^2 > 50\%$ and (or) P < 0.1, the random-effects model was applied; otherwise, the fix-effects model was applied. The sensitivity analysis was performed to evaluate stability of results. Furthermore, Begg's funnel plot and Egger's test were conducted to detect publication bias [17, 18].

# Results

## Literature search

A total of 126 records were identified from databases and 27 duplicated records were removed. Eventually, nine studies were included [19–27]. The detailed selection process was shown in the **Fig 1**.

## Basic characteristics of included studies

Among nine retrospective included studies, 1476 patients were enrolled and the sample size ranged from 80 to 500 [19–27]. Among four of included studies the cutoff values of SMI, 55cm$^2$/m$^2$ for male and 39cm$^2$/m$^2$ for female, were applied [19, 21, 24, 27]. In the other five studies, the cutoff values of SMI, 43/53cm$^2$/m$^2$ for male and 39cm$^2$/m$^2$ for female, were applied and the cutoff value of SMI for male patients was adjusted by the body mass index (BMI), SMI <43cm$^2$/m$^2$ for patients with BMI<25kg/m$^2$ and <53cm$^2$/m$^2$ for patients with BMI≥25 kg/m$^2$ [20, 22, 23, 25, 26]. All included studies were with high-quality with a NOS score ≥6. Specific information was displayed in **Table 1**.

## The predictive role of pretreatment SMI for OS in bladder cancer

All included studies explored predictive role of pretreatment SMI for OS [19–27]. Pooled results indicated that a lower pretreatment SMI was significantly associated with poor OS in bladder cancer (HR = 1.56, 95% CI: 1.33–1.82, P<0.001; $I^2$ = 6.4%, P = 0.382) (**Fig 2**). Subgroup analysis stratified by thresholds of SMI showed similar results (non-adjusted SMI:

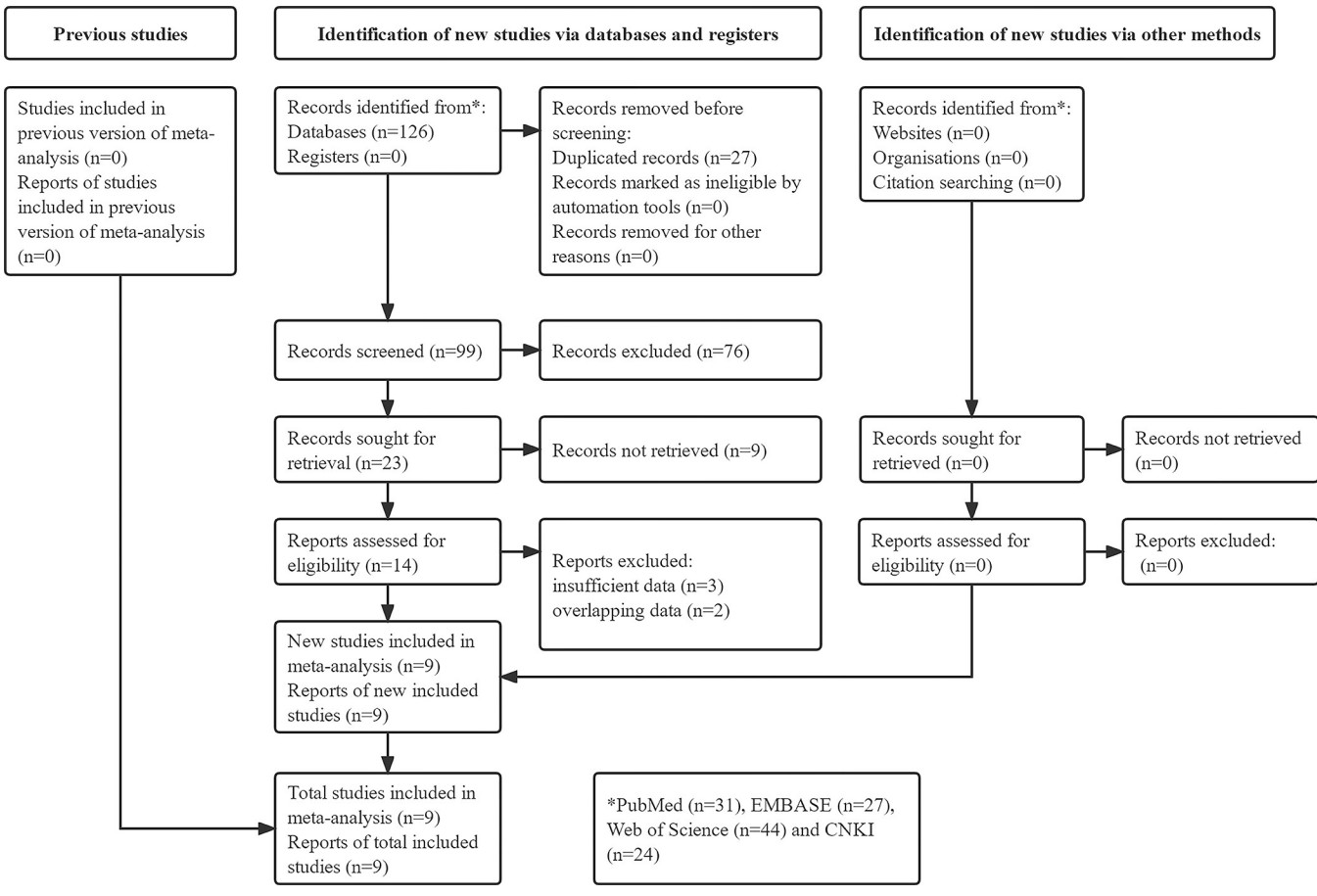

**Fig 1. Prisma flow diagram of this meta-analysis.**

**Table 1. Basic characteristics of included studies.**

| Author | Year | Country | Sample size | TNM stage | Treatment | Cutoff value | Endpoint | NOS |
|---|---|---|---|---|---|---|---|---|
| Psutka [19] | 2014 | USA | 205 | Mixed | Surgery | male: $55cm^2/m^2$, female: $39cm^2/m^2$ | OS, CSS | 7 |
| Miyake [20] | 2017 | Japan | 89 | NR | Surgery | male: $43/53cm^2/m^2$, female: $41cm^2/m^2$ | OS, CSS | 7 |
| Abe [21] | 2018 | Japan | 87 | NR | Chemotherapy/ chemotherapy plus surgery | male: $55cm^2/m^2$, female: $39cm^2/m^2$ | OS | 6 |
| Mayr [22] | 2018 | Netherlands | 500 | Mixed | Surgery | male: $43/53cm^2/m^2$, female: $41cm^2/m^2$ | OS, CSS | 7 |
| Ha [23] | 2019 | Republic of Korea | 80 | Mixed | Surgery | male: $43/53cm^2/m^2$, female: $41cm^2/m^2$ | OS | 7 |
| Lyon [24] | 2019 | USA | 183 | Mixed | Surgery | male: $55cm^2/m^2$, female: $39cm^2/m^2$ | OS, CSS | 7 |
| Stangl [25] | 2019 | Austria | 94 | Mixed | Radiotherapy | male: $43/53cm^2/m^2$, female: $41cm^2/m^2$ | OS, CSS | 6 |
| Yuan [26] | 2021 | China | 97 | cT1-2 | Surgery | male: $43/53cm^2/m^2$, female: $41cm^2/m^2$ | OS | 8 |
| Almarzouq [27] | 2022 | Canada | 141 | Mixed | Radiotherapy plus chemotherapy | male: $55cm^2/m^2$, female: $39cm^2/m^2$ | OS | 6 |

NR: not reported; OS: overall survival; CSS: cancer-specific survival; NOS: Newcastle-Ottawa Scale.

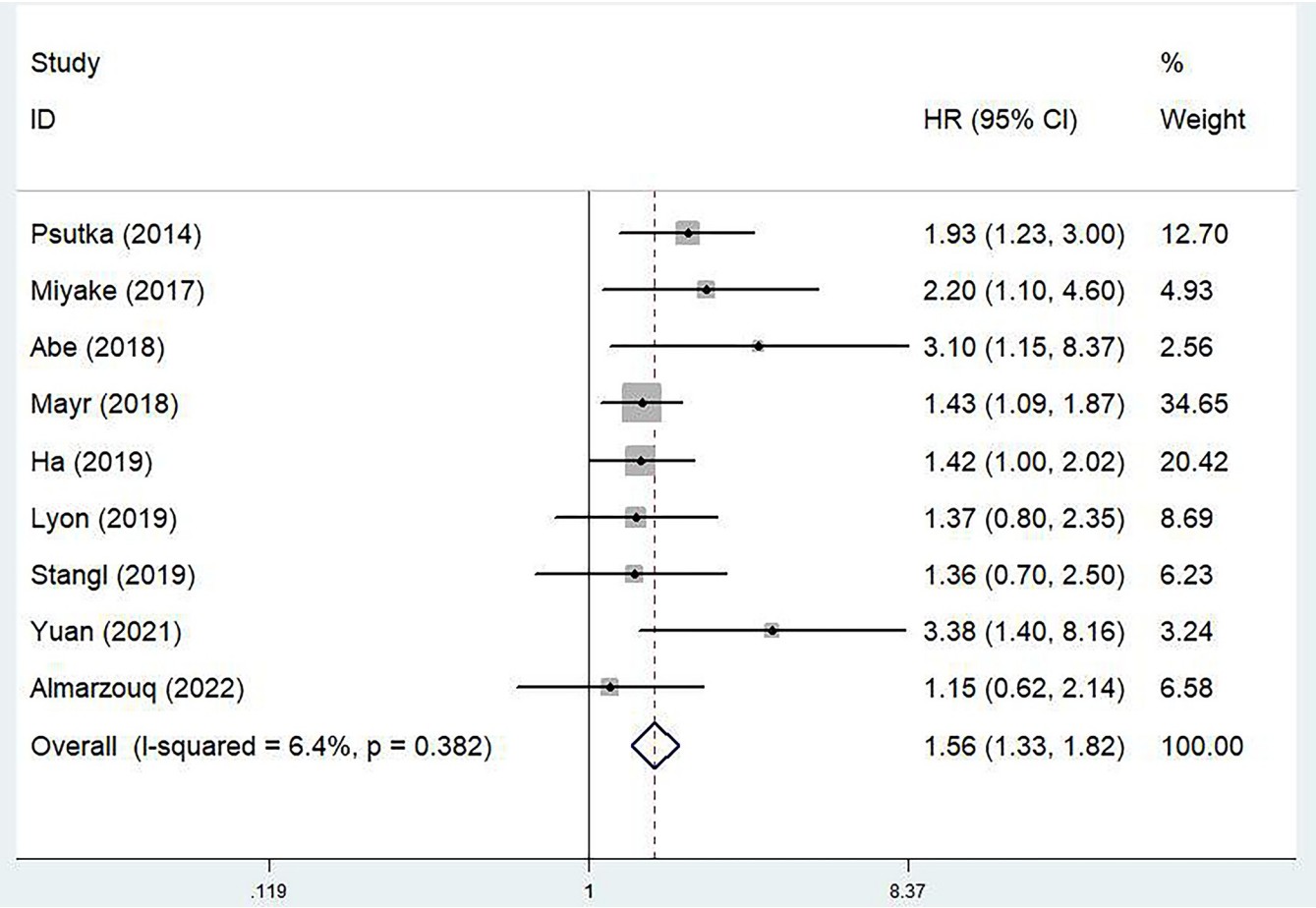

**Fig 2. The association between pretreatment skeletal muscle mass index and overall survival of bladder cancer patients.**

HR = 1.63, 95% CI: 1.22–2.17, P = 0.001; BMI-adjusted: HR = 1.52, 95% CI: 1.26–1.84, P<0.001) (Table 2).

## The predictive role of pretreatment SMI for CSS in bladder cancer

Five studies investigated predictive role of SMI for CSS of bladder cancer patients [19, 20, 22, 24, 25]. Pooled results revealed that lower pretreatment SMI was related to worse CSS (HR = 1.75, 95% CI: 1.36–2.25, P<0.001; $I^2$ = 15.5%, P = 0.316) (Fig 3).

**Table 2. Results of meta-analysis.**

|  | No. of studies | HR | 95% CI | P value | $I^2$ (%) | P value |
|---|---|---|---|---|---|---|
| Overall survival | 9 [19–27] | 1.56 | 1.33–1.82 | <0.001 | 6.4 | 0.382 |
| Cutoff value of SMI |  |  |  |  |  |  |
| Non-adjusted | 4 [19, 21, 24, 27] | 1.63 | 1.22–2.17 | 0.001 | 20.7 | 0.286 |
| BMI-adjusted | 5 [20, 22, 23, 25, 26] | 1.52 | 1.26–1.84 | <0.001 | 13.4 | 0.329 |
| Cancer-specific survival | 5 [19, 20, 22, 24, 25] | 1.75 | 1.36–2.25 | <0.001 | 15.5 | 0.316 |

HR: hazard ratio; CI: confidence interval; SMI: skeletal muscle mass index.

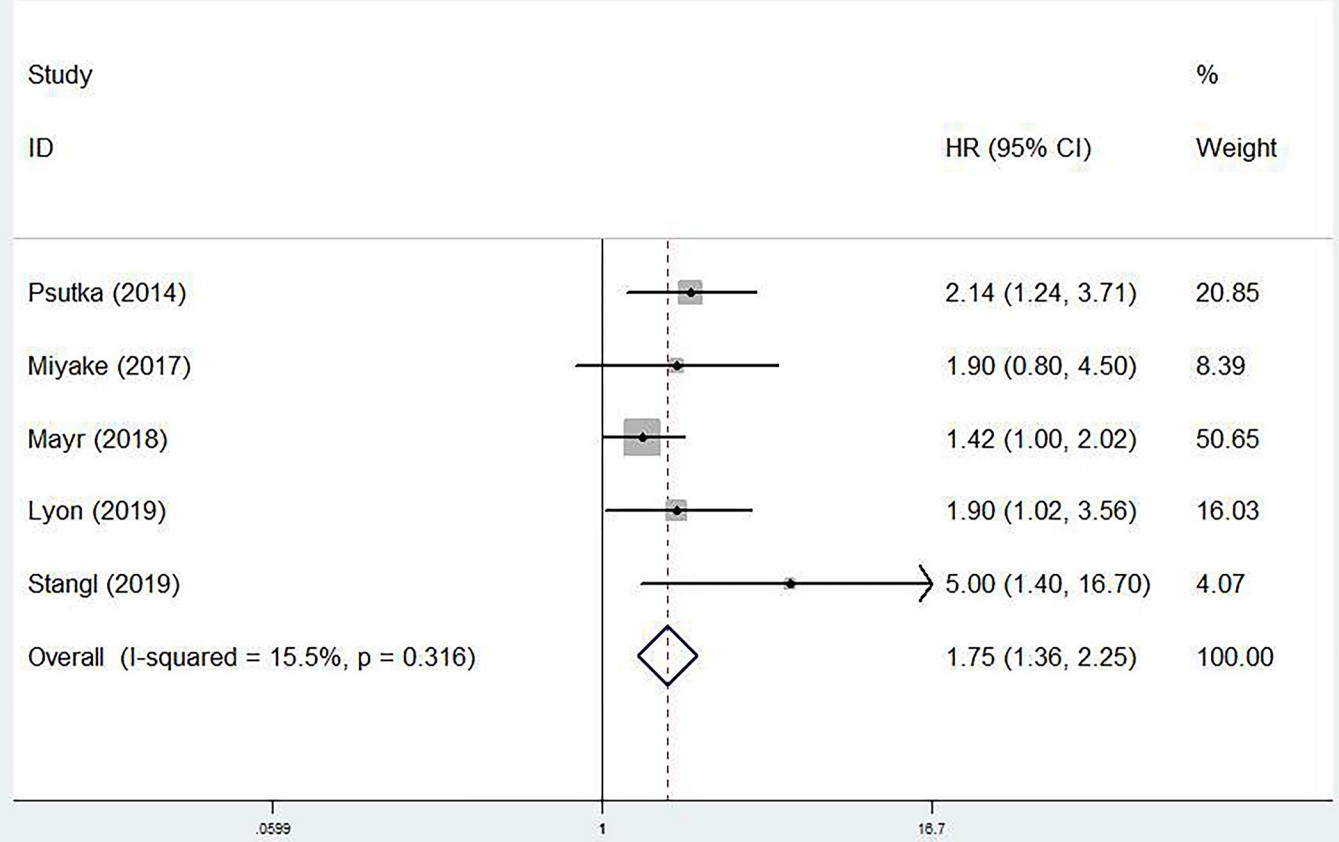

**Fig 3. The association between pretreatment skeletal muscle mass index and cancer-specific survival of bladder cancer patients.**

### Sensitivity analysis and publication bias

Sensitivity analysis for OS demonstrated that our results were stable and reliable (**Fig 4**). Besides, symmetrical Begg's funnel plot (**Fig 5**) and P = 0.096 of Egger's test both indicated non-significant publication bias.

## Discussion

The current meta-analysis demonstrated that pretreatment SMI was associated with long-term survival in bladder cancer and lower pretreatment SMI predicted poorer OS and CSS. Therefore, pretreatment SMI might serve as a reliable prognostic indicator in bladder cancer. However, due the limitations existed in this meta-analysis like to the retrospective nature of included studies more prospective high-quality studies are still needed to verify our results.

Actually, the predictive role of SMI for survival in cancers has been verified. Pan et al. included 12 studies involving 3002 cases and demonstrated that a lower SMI was obviously related to poorer OS (HR = 1.23, P<0.001) [12]. The subgroup analysis based on treatment, stage and tumor type further manifested prognostic role of SMI in lung cancer and showed similar results [12]. Yao et al. enrolled 2441 patients from 17 studies and demonstrated that lower pretreatment SMI was associated with poorer OS (HR = 1.18, P<0.001) and disease-free survival (DFS) (HR = 1.78, P = 0.019) [13]. Subgroup analysis stratified by the treatment, tumor type and thresholds of SMI revealed similar findings [13]. Our meta-analysis was the first to determine predictive role of pretreatment SMI in bladder cancer and strongly verified that lower pretreatment SMI was related to worse prognosis.

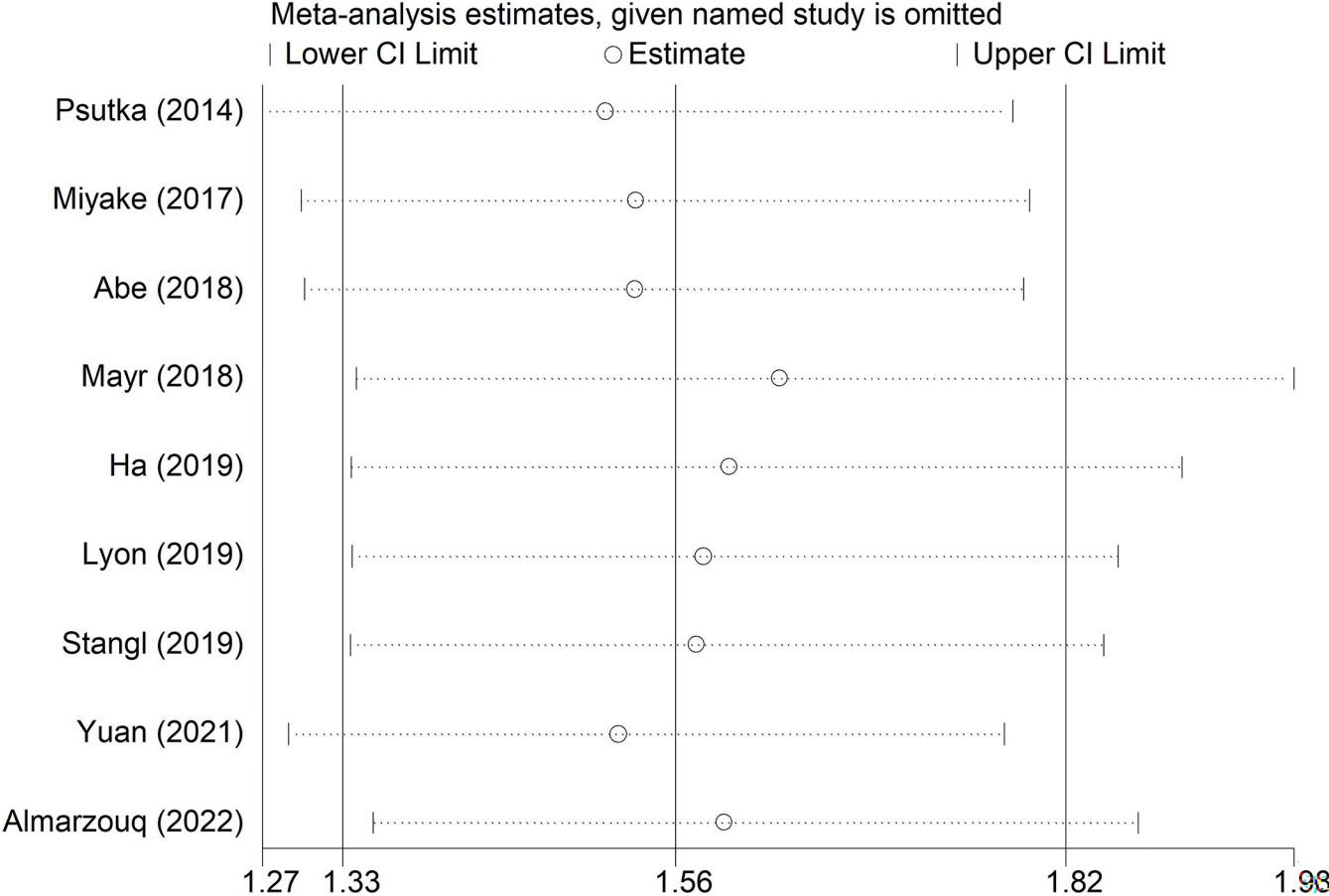

**Fig 4. Sensitivity analysis about the association between pretreatment skeletal muscle mass index and overall survival of bladder cancer patients.**

SMI is the most common indicator assessing the presence or absence of sarcopenia. Initially, sarcopenia is regarded as a disease of old age characterized by degeneration of muscle tissue. However, increasing evidence indicated that a number of factors could cause sarcopenia such as the disuse, cachexia, malabsorption and also tumors [28]. Meanwhile, the occurrence and development of sarcopenia are closely related to the prognosis of cancer patients [13]. Sarcopenia includes physiological and pathological sarcopenia and the latter is caused by malignant or benign diseases. Tumor-associated sarcopenia is usually closely related to cachexia, representing as marked muscle mass loss and systemic chronic inflammation [29, 30]. The incidence rate of tumor-associated sarcopenia is about 50%-90% in untreated cancer patients [31]. Among patients with bladder cancer, the occurrence rate of sarcopenia is more than 50% [32]. Bladder cancer patients may experience malnutrition due to the impact of the tumor on the body's metabolism and absorption, or due to adverse reactions during treatment, such as loss of appetite, nausea, and vomiting. Besides, bladder cancer patients with sarcopenia may have reduced tolerance to surgical and chemotherapeutic treatments, and the disruption of the body's immune and metabolic functions may interfere with the normal response to these treatments [33]. In the past years, the association between sarcopenia and prognosis in cancers has been widely reported and revealed. For now, the predictive role of sarcopenia has been confirmed in several types of tumors including esophageal cancer, rectal cancer and hepatocellular carcinoma [34–37]. Therefore, our meta-analysis indirectly proved that the sarcopenia

Begg's funnel plot with pseudo 95% confidence limits

**Fig 5. Begg's funnel plot.**

assessed by SMI before any anti-tumor treatment was a novel and reliable prognostic factor in bladder cancer.

In the current meta-analysis, we failed to conduct more analysis about clinical role of SMI in bladder cancer because of lack of original data and limited current evidence. There are still many fields worthy of further investigations. For example, our meta-analysis only identified the association between pretreatment SMI and long-term survival. However, whether the change of SMI during the anti-tumor treatment could predict survival and contribute to the therapy strategy remains unclear. Besides, the cutoff values of SMI are gender-specific and BMI is sometimes considered. It is not clear whether more parameters should be considered such as the age and tumor stage. Furthermore, skeletal muscle plays an essential role in the systemic inflammation response and a large number of evidences have shown that the status of systemic inflammation response is closely related to prognosis of cancer patients [38–40]. Thus, a combination of SMI and some inflammation indexes like the SII might be better in predicting the long-term survival of bladder cancer patients.

## Limitation of this study

Several limitations exist in this meta-analysis. First, all included studies are retrospective with relatively small sample sizes, which might cause some bias. Second, some clinicopathological parameters are unobtainable such as the TNM stage and age and we were unable to conduct more subgroup analysis based on these important indicators due to the lack of original data.

## Conclusion

Lower pretreatment SMI was associated with worse long-term survival of bladder cancer patients. However, more prospective high-quality studies are still needed to verify our results.

## Supporting information

**S1 File. PRISMA 2020 checklist for this meta-analysis.**
(DOCX)

## Author Contributions

**Conceptualization:** Jingjing An.

**Data curation:** Qian Yuan, Jianrong Hu, Feng Yuan.

**Formal analysis:** Qian Yuan, Feng Yuan.

**Investigation:** Qian Yuan.

**Methodology:** Qian Yuan, Jianrong Hu, Feng Yuan.

**Resources:** Feng Yuan.

**Software:** Jianrong Hu, Feng Yuan.

**Supervision:** Jingjing An.

**Validation:** Jianrong Hu.

**Writing – original draft:** Qian Yuan, Feng Yuan.

**Writing – review & editing:** Jianrong Hu, Jingjing An.

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
