## [Decision Letter · Decision Letter 0]

16 Apr 2023

PONE-D-23-02766Predictive role of pretreatment skeletal muscle mass index for long-term survival of bladder cancer patients: a meta-analysisPLOS ONE

Dear Dr. an,

Thank you for submitting your manuscript to PLOS ONE. After careful consideration, we feel that it has merit but does not fully meet PLOS ONE’s publication criteria as it currently stands. Therefore, we invite you to submit a revised version of the manuscript that addresses the points raised during the review process.

We look forward to receiving your revised manuscript.

Kind regards,

Andrea D’Aviero

Academic Editor

PLOS ONE

Journal Requirements:

2. Thank you for submitting the above manuscript to PLOS ONE. During our internal evaluation of the manuscript, we found significant text overlap between your submission and previous work in the abstract, methods, results and discussion.

Please revise the manuscript to rephrase the duplicated text, cite your sources, and provide details as to how the current manuscript advances on previous work. Please note that further consideration is dependent on the submission of a manuscript that addresses these concerns about the overlap in text with published work.

We will carefully review your manuscript upon resubmission and further consideration of the manuscript is dependent on the text overlap being addressed in full. Please ensure that your revision is thorough as failure to address the concerns to our satisfaction may result in your submission not being considered further.

Additional Editor Comments:

The authors present a paper about a topic of interest but major revisions are needed as suggested by reviewers.

Reviewers' comments:

Reviewer's Responses to Questions

**Comments to the Author**

1. Is the manuscript technically sound, and do the data support the conclusions?

Reviewer #1: Yes

Reviewer #2: Yes

2. Has the statistical analysis been performed appropriately and rigorously? 

Reviewer #1: I Don't Know

Reviewer #2: Yes

3. Have the authors made all data underlying the findings in their manuscript fully available?

Reviewer #1: Yes

Reviewer #2: Yes

4. Is the manuscript presented in an intelligible fashion and written in standard English?

Reviewer #1: Yes

Reviewer #2: Yes

5. Review Comments to the Author

Reviewer #1: The authors present a paper about "Predictive role of pretreatment skeletal muscle mass index for long-term survival of

bladder cancer patients: a meta-analysis".

The topic is absolutely interesting but there are a few points that I would like the authors to address more in detail as follows:

1) Please explain the rationale to focus on pretreatment skeletal muscle mass index as indicator of long-term survival: is it just for sarcopenia? is there a specific link with bladder cancer? is it related to the treatment burden (chemotherapy? surgery? radiotherapy?)

2) Is the value of pretreatment skeletal muscle mass index prognostic or predictive?

3) Please add further information about the "non-surgery" and "mixed" treatment to table 1

Reviewer #2: The paper reports an interesting systematic review with pooled analysis on the role of the pretreatment skeletal muscle mass index in predicting survival outcomes.

The introduction well circumscribes the study in the scientific landscape, the rigorous and appropriate methodologies are clearly stated, the results are comprehensively presented and the discussion offers perspectives and limitations of the work.

The language is clear and needs no further revision.

Translated with www.DeepL.com/Translator (free version)

6. PLOS authors have the option to publish the peer review history of their article (what does this mean?). If published, this will include your full peer review and any attached files.

Reviewer #1: No

Reviewer #2: **Yes: **Calogero Casà

---

## [Author Response · Author response to Decision Letter 0]

18 Apr 2023

Response to journal requirements: 

Answer 1: We have thoroughly and carefully checked and modified this manuscript according to PLOS ONE’s style requirements.

2. Thank you for submitting the above manuscript to PLOS ONE. During our internal evaluation of the manuscript, we found significant text overlap between your submission and previous work in the abstract, methods, results and discussion.

Please revise the manuscript to rephrase the duplicated text, cite your sources, and provide details as to how the current manuscript advances on previous work. Please note that further consideration is dependent on the submission of a manuscript that addresses these concerns about the overlap in text with published work.

We will carefully review your manuscript upon resubmission and further consideration of the manuscript is dependent on the text overlap being addressed in full. Please ensure that your revision is thorough as failure to address the concerns to our satisfaction may result in your submission not being considered further.

Answer 2: Dear editor, thanks for your comment. We have completely revised our manuscript to reduce the repetition rate, from 40% to 30%. However, the duplication is mainly in the methodological section. The similarity detection was performed by Turnitin system. If necessary, we would be happy to further reduce the repetition rate.

Answer 3: We have added the caption for the Supporting Information file at the end of this manuscript (page 24. line 1) and also cited this file in the text (page 4, line 7-8)

Response to reviewer 1:

Reviewer #1: The authors present a paper about "Predictive role of pretreatment skeletal muscle mass index for long-term survival of bladder cancer patients: a meta-analysis".

The topic is absolutely interesting but there are a few points that I would like the authors to address more in detail as follows:

Question 1: Please explain the rationale to focus on pretreatment skeletal muscle mass index as indicator of long-term survival: is it just for sarcopenia? is there a specific link with bladder cancer? is it related to the treatment burden (chemotherapy? surgery? radiotherapy?)

Answer 1: Dear reviewer, thanks for your valuable comment. We have carefully explained the rationale to focus on pretreatment SMI as indicator of long-term survival in the discussion part. 

“SMI is the most common indicator assessing the presence or absence of sarcopenia. Initially, sarcopenia is regarded as a disease of old age characterized by degeneration of muscle tissue. However, increasing evidence indicated that a number of factors could cause sarcopenia such as the disuse, cachexia, malabsorption and also tumors [28]. Meanwhile, the occurrence and development of sarcopenia are closely related to the prognosis of cancer patients [13]. Sarcopenia includes physiological and pathological sarcopenia and the latter is caused by malignant or benign diseases. Tumor-associated sarcopenia is usually closely related to cachexia, representing as marked muscle mass loss and systemic chronic inflammation [29, 30]. The incidence rate of tumor-associated sarcopenia is about 50%-90% in untreated cancer patients [31]. Among patients with bladder cancer, the occurrence rate of sarcopenia is more than 50% [32]. Bladder cancer patients may experience malnutrition due to the impact of the tumor on the body's metabolism and absorption, or due to adverse reactions during treatment, such as loss of appetite, nausea, and vomiting. Besides, bladder cancer patients with sarcopenia may have reduced tolerance to surgical and chemotherapeutic treatments, and the disruption of the body's immune and metabolic functions may interfere with the normal response to these treatments [33]. In the past years, the association between sarcopenia and prognosis of cancer patients has been widely reported and revealed. For now, the predictive role of sarcopenia has been verified in several types of cancers such as the esophageal cancer, rectal cancer and hepatocellular carcinoma [34-37]. Therefore, our meta-analysis indirectly proved that the sarcopenia assessed by SMI before any anti-tumor treatment was a novel and reliable prognostic factor in bladder cancer.” (page 12, line 17-22; page 13, line 1-17)

Question 2: Is the value of pretreatment skeletal muscle mass index prognostic or predictive?

Answer 2: Dear reviewer, we deem that these two phrases, “predictive role of SMI for survival” and “prognostic role of SMI”, may mean the same thing. After carefully reviewing previous similar articles, we found that both of these expressions were quite common although the latter is more common. If necessary, we would like to unify to the latter type of expression, “prognostic”.

Question 3: Please add further information about the "non-surgery" and "mixed" treatment to table 1

Answer 3: We have added further information about the “non-surgery” and “mixed” treatment in table 1. (page 7-8)

Response to reviewer 2

The paper reports an interesting systematic review with pooled analysis on the role of the pretreatment skeletal muscle mass index in predicting survival outcomes.

The introduction well circumscribes the study in the scientific landscape, the rigorous and appropriate methodologies are clearly stated, the results are comprehensively presented and the discussion offers perspectives and limitations of the work.

The language is clear and needs no further revision.

Answer: Thank you very much for your recognition of the quality of this article

---

## [Decision Letter · Decision Letter 1]

19 Jun 2023

Predictive role of pretreatment skeletal muscle mass index for long-term survival of bladder cancer patients: a meta-analysis

PONE-D-23-02766R1

Dear Dr. an,

We’re pleased to inform you that your manuscript has been judged scientifically suitable for publication and will be formally accepted for publication once it meets all outstanding technical requirements.

Kind regards,

Andrea D’Aviero

Academic Editor

PLOS ONE

Additional Editor Comments (optional):

The authors have satisfactorily addressed all the reviewer comments.

Reviewers' comments:

Reviewer's Responses to Questions

**Comments to the Author**

1. If the authors have adequately addressed your comments raised in a previous round of review and you feel that this manuscript is now acceptable for publication, you may indicate that here to bypass the “Comments to the Author” section, enter your conflict of interest statement in the “Confidential to Editor” section, and submit your "Accept" recommendation.

Reviewer #1: All comments have been addressed

2. Is the manuscript technically sound, and do the data support the conclusions?

Reviewer #1: Yes

3. Has the statistical analysis been performed appropriately and rigorously? 

Reviewer #1: N/A

4. Have the authors made all data underlying the findings in their manuscript fully available?

Reviewer #1: Yes

5. Is the manuscript presented in an intelligible fashion and written in standard English?

Reviewer #1: Yes

6. Review Comments to the Author

Reviewer #1: I have no further comments, the authors have satisfactorily addressed all of my previous comments in the responses provided

7. PLOS authors have the option to publish the peer review history of their article (what does this mean?). If published, this will include your full peer review and any attached files.

Reviewer #1: No

---

## [Editor Report · Acceptance letter]

22 Jun 2023

PONE-D-23-02766R1 

Predictive role of pretreatment skeletal muscle mass index for long-term survival of bladder cancer patients: a meta-analysis 

Dear Dr. An:

I'm pleased to inform you that your manuscript has been deemed suitable for publication in PLOS ONE. Congratulations! Your manuscript is now with our production department. 

Kind regards, 

on behalf of

Dr. Andrea D’Aviero 

Academic Editor

PLOS ONE